# The respiratory microbiome in pulmonary tuberculosis: a meta-analysis reveals niche-specific microbial and functional signatures

Mingyang Qin,[1,2] Yanhua Wen,[3] Shanshan Li,[1] Song Li,[3] Xuming Li,[3] Yuting Lin,[3] Long Hu,[3] Han Xia,[3] Yu Pang,[2] Liang Li[1,2]

**ABSTRACT** Tuberculosis (TB) remains a major global health challenge. The close relationship between the microbiome and the host is becoming increasingly notable. While studies on the respiratory microbiome in pulmonary tuberculosis (PTB) exist, a comprehensive understanding of microbial characteristics across the entire respiratory tract is still lacking. To address this, we conducted a meta-analysis by integrating data from common and representative respiratory samples. We integrated 16S rRNA data from 11 public datasets encompassing upper respiratory tract specimens (URTs), sputum, and bronchoalveolar lavage fluid (BALF). Ecological patterns were investigated through co-occurrence networks and neutral community modeling, while taxonomic and functional analyses were conducted with QIIME2 and PICRUSt2. The respiratory microbiota in PTB exhibited dynamic variations while sharing common genera, such as *Streptococcus*, *Prevotella*, *Veillonella*, and *Neisseria*. Alpha diversity was consistently higher in PTB than in healthy controls, with BALF exhibiting the greatest microbial diversity. Several differentially abundant genera were identified among the three sample types, *Serratia* being almost exclusively detected in BALF. Notably, the microbial interaction network in sputum was more complex and demonstrated the best fit to the neutral community model. Functional predictions highlighted enriched pathways such as peptidoglycan maturation and ABC transporters, and *Bacillus* was linked to multiple metabolic pathways. Several KO functions were predicted to be more active in URTs and sputum than in BALF. Our multi-scale analysis delineates a niche-specific biogeography of the respiratory microbiome in PTB. By elucidating community assembly and microbe interplay, we position the respiratory microbiota as an active contributor to TB. This work paves the way for novel microbiota-based diagnostics and ecologically informed therapies.

**IMPORTANCE** Pulmonary tuberculosis (PTB) remains a leading cause of global mortality, yet the ecological principles shaping its respiratory microbiome are poorly understood. By integrating 16S rRNA datasets from upper and lower airway specimens, this study provides the first comprehensive meta-analysis of respiratory microbial diversity and function in PTB. We reveal distinct community structures and functional potentials among upper airways, sputum, and bronchoalveolar lavage fluid, driven by niche-specific ecological processes rather than stochastic assembly. These findings establish a baseline framework for interpreting microbial biogeography across the respiratory tract and highlight potential microbial biomarkers for site-specific monitoring and therapeutic targeting in PTB.

**KEYWORDS** tuberculosis, 16S, microbiome, respiratory tract, meta

**Peer Reviewer** Simbarashe Peter Zvada, University of Cape Town, Cape Town, South Africa

Address correspondence to Liang Li, Liliang69@vip.sina.com, Yu Pang, pangyupound@163.com, or Han Xia, xiahan@hugobiotech.com.

Mingyang Qin, Yanhua Wen, Shanshan Li, and Song Li contributed equally to this article. Author order was determined by drawing straws.

The authors declare no conflict of interest.

Tuberculosis (TB), caused by *Mycobacterium tuberculosis* (MTB), remains a devastating global health threat (1). Although the tuberculosis pathogenesis has traditionally focused on the host immune response (2) and the virulence of MTB (3), increasing evidence has revealed that the host-associated microbiota also plays a crucial role in immune regulation, pathogen colonization resistance, and disease progression (4). In recent years, advances in high-throughput sequencing technologies have enabled a deeper understanding of the complex microbial ecology associated with respiratory diseases (5, 6).

The composition of bacterial populations residing in the human body varies across different anatomical sites (7). Mounting studies indicated that the resident microbiome of the respiratory tract can modulate host immunity, influence pathogen colonization, and ultimately shape disease outcomes (8, 9). The respiratory tract is a continuous yet compartmentalized ecosystem, in which different regions of the upper and lower airways are closely connected both physiologically and ecologically. Microorganisms may migrate between these regions through inhalation, mucociliary clearance, or local immune responses, thereby forming a dynamic symbiotic network (9, 10). However, current research predominantly focuses on single sample types, and due to its non-invasive accessibility, most studies are heavily reliant on sputum samples.

Moreover, in pulmonary tuberculosis (PTB) research, distinct sampling strategies for microbiome analysis may yield differing responses to MTB infection. Upper respiratory tract specimens (URTs) are relatively easy to obtain and have been shown to reflect lung communities and disease progression in other respiratory diseases (11, 12). Routinely collected sputum is commonly used for TB diagnosis but carries the risk of contamination by the upper respiratory tract (URT) microbiota (13). Bronchoalveolar lavage fluid (BALF) is considered the gold standard for sampling the lower airways; however, its collection is invasive and consequently less studied in TB microbiome research. It remains unknown whether these three specimen types yield consistent or divergent microbial signatures in PTB. This has resulted in a lack of systematic understanding regarding the overall differences and interrelationships of microbial communities across different regions of the respiratory tract in patients with PTB.

To address this gap, we performed a meta-analysis integrating microbiome data from URTs, sputum, and BALF collected from patients with PTB. By systematically synthesizing microbial profiles from multiple respiratory sites, our study aims to elucidate the common and site-specific microbial signatures associated with TB, identify potential microbial biomarkers, and explore the ecological patterns underlying respiratory microbial dysbiosis during infection. This integrated analysis provides novel insights into the spatial heterogeneity of the respiratory microbiome in TB and contributes to a better understanding of the host–microbe interactions involved in TB pathogenesis.

## MATERIALS AND METHODS

### Research inclusion and screening process

We conducted a systematic literature search on PubMed and Web of Science using the keywords "tuberculosis," "respiratory tract," and "microbiome/microbiota" to identify studies investigating the respiratory microbiome in PTB patients. Studies were systematically identified and selected in strict adherence to the Preferred Reporting Items for Systematic Reviews and Meta-Analyses (PRISMA) guidelines (Fig. 1). We retrieved raw sequencing data using project numbers provided in the publications. Furthermore, we expanded our data collection by searching multiple public databases for relevant data sets. Based on our inclusion criteria, a total of 11 data sets were ultimately included in this meta-analysis, with detailed information summarized in the table (Table 1). Additionally, the controls across the 11 data sets were heterogeneous, including nontuberculosis groups, latent TB infection groups, household contacts of TB patients, and healthy control groups. Therefore, only samples definitively classified as healthy controls (HC) were included in the subsequent comparative study (Table S1).

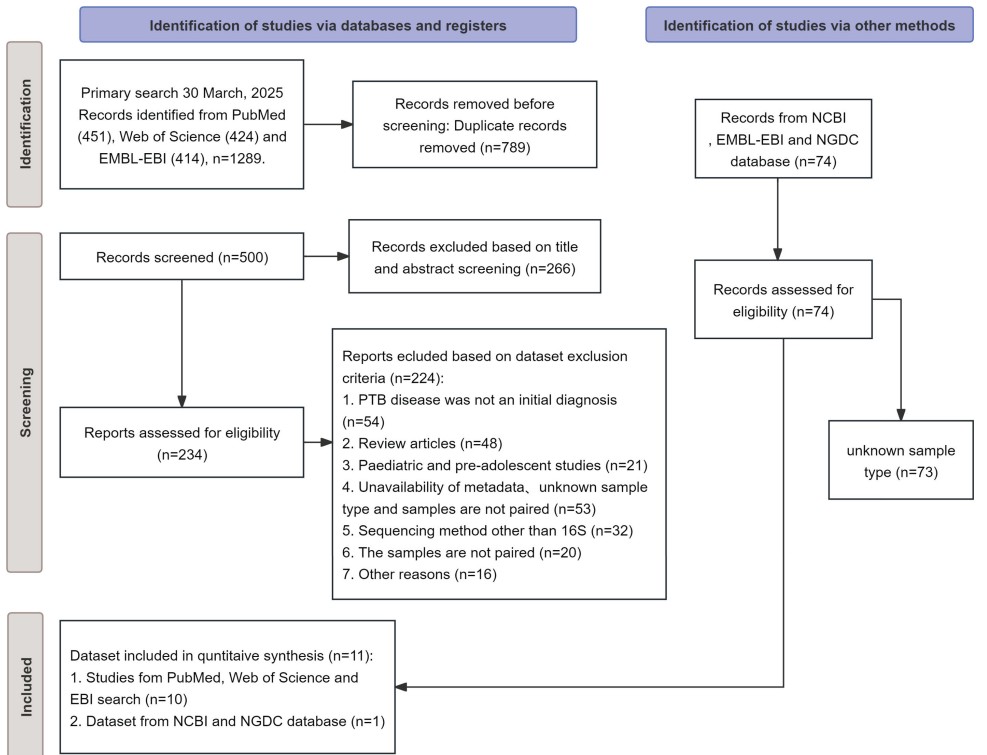

**FIG 1** PRISMA flow diagram illustrating the study selection process for the study. The diagram outlines the identification, screening, eligibility, and inclusion stages of studies retrieved from databases (PubMed, Web of Science, and EMBL-EBI) and other sources (NCBI, NGDC). Following the primary search on 30 March 2025, 1,289 records were identified. After the removal of 789 duplicates, 500 records were screened based on title and abstract, leading to the exclusion of 266 records. The remaining 234 full-text records were assessed for eligibility, with 223 records excluded for reasons including incorrect initial diagnosis, being review articles, pediatric studies, unavailability of paired metadata, non-16S sequencing methods, and other criteria. An additional 74 records from databases were assessed, of which 73 were excluded due to an unknown sample type. Ultimately, 11 records (10 from database searches and 1 data set from public databases) were included in the qualitative synthesis.

## Data processing for taxonomic annotation and functional analysis

Publicly obtained raw reads were processed in QIIME2 (14) (version 2022.2) using the following workflow. Primer sequences were trimmed using the cutadapt plugin.

**TABLE 1** The detailed information of the original data included in the analysis

| Year | Data set | Platform | Data type | Site | Sample type (*n*) | Sample origin |
|---|---|---|---|---|---|---|
| 2014 | PRJNA242354 | 454 | V1-V2 | URT | Nasal (6), oropharynx (6), sputum (6) | Columbia |
| 2018 | PRJNA432583 | MiSeq | V3-V4 | URT | Nasopharyngeal and oropharyngeal swabs (6) | Myanmar |
| 2020 | PRJNA580164 | Ion S5 | V3-V4 | Lung | BALF (6) | China |
| 2020 | PRJNA611472 | MiSeq | V1-V2 | Airway | Sputum (58) | South Africa |
| 2021 | PRJNA663902 | 16S | V4 | Airway | Sputum (70) | Ethiopia |
| 2021 | PRJNA664352 | MiSeq | V4 | URT&Airway | Oral wash (58) sputum (55) | South Africa |
| 2022 | PRJNA824137 | MiSeq | V3-V4 | Lung | BALF (23) | China |
| 2022 | PRJNA867528 | MiSeq | V3-V4 | Lung | BALF (21) | China |
| 2022 | PRJNA837186 | HiSeq | V1-V3 | Lung | Sputum (30) | China |
| 2024 | https://doi.org/10.5281/zenodo.11560386 | MiSeq | V3-V4 | Airway | Sputum (50) | India |
| 2025 | PRJNA1103672 | MiSeq | V3-V4 | Lung | BALF (23) | Germany |

Denoising, paired-end sequence merging, and chimera removal were performed using the dada2 (15) method, resulting in a final set of high-quality, non-chimeric sequences. Amplicon sequence variants (ASVs) were inferred from these sequences using default parameters. Taxonomic assignment of ASVs was carried out with the classify-sklearn module in QIIME2. A pre-trained SILVA (16) v138 99% OTUs full-length Naive Bayes classifier was used for this purpose, with a confidence threshold set to 0.7. Resulting ASVs classified as chloroplasts or mitochondria were filtered out. To normalize sequencing depth, all samples were rarefied to 10,000 reads per sample. Finally, genus-level abundance tables were generated for downstream analysis.

Microbial metabolic functions were predicted using PICRUSt2 (v2.6.2) (17) to infer the relative abundances of Kyoto Encyclopedia of Genes and Genomes (KEGG) Orthologs (KOs) and MetaCyc metabolic pathways. The resulting KO and pathway abundance tables were normalized across all samples. Associations between microbial genera and metabolic pathways were assessed using Spearman's rank correlation, but only for pathway-genus pairs where the metabolic pathway is documented to exist within that genus or its closely related species according to the MetaCyc database (18).

## Correction of technical batch effects

To address batch effects between data sets, we performed batch correction using the R package ConQuR (v1.2.0) (19). ConQuR was applied to genus-level read count data, with the data set source specified as the batch variable and disease state and sample type included as covariates. The correction utilized a penalized fitting strategy with the following parameters: logistic_lasso = T, quantile_type = "lasso," and interplt = T (20). Then, for correcting batch effects in the functional profiles of the samples, we employed the ComBat-Seq algorithm (R package sva) (21). ComBat-Seq was run using a full model, where the data set source was defined as the batch variable, disease state as the group variable, and sample type as a covariate.

## Interaction network

Microbial co-occurrence networks were constructed separately for the URTs, sputum, and BALF groups using the SparCC (22) algorithm. Only genera detected in ≥20% of samples within each group were retained for network inference, based on their read-count data. Pairwise SparCC correlations were estimated over 100 iterative runs. To evaluate statistical significance, 1,000 randomly permuted data sets were generated, and correlations were recomputed for each. The $P$-value for each observed correlation was derived empirically by determining the proportion of random correlations that were stronger than the original. For network visualization, only correlations satisfying $| r | >$ 0.2 and a permutation-based $P < 0.05$ were retained and imported into Gephi software (https://doi.org/10.1609/icwsm.v3i1.13937).

## Neutral model

The role of neutral processes in the assembly of respiratory microbiota in PTB patients was evaluated by fitting a neutral community model (23). The model was fitted using a customized R script, with URTs, sputum, and BALF samples serving as the respective microbial source communities. Parameter fitting was performed via nonlinear least squares regression using the minpack.lm package in R. The model fit was evaluated by the coefficient of determination ($R^2$), and the value ranged from ≤ 0 (no fit) to 1 (perfect fit). Then, 95% confidence intervals for predictions were calculated using the Wilson score method (Hmisc package).

## Statistical analysis

All statistical analyses were performed in R (version 4.3.1) within RStudio and using ggplot2 for visualization. Alpha diversity (Shannon, Simpson, Chao1, and ACE) and beta diversity (principal coordinates analysis [PCoA]) were computed with the vegan package at genus level. Differential taxa were identified through a stepwise analysis: significant features across groups from the Kruskal-Wallis test were further confirmed by pairwise Wilcoxon tests, and their effect sizes were quantified by linear discriminant analysis (LDA) effect size (LEfSe) (LDA > 3) (24, 25). Statistical significance for inter-group comparisons was determined using the Wilcoxon rank-sum test (for two groups) or the Kruskal-Wallis (KW) test (for multiple groups), with a significance threshold of $P < 0.05$.

## RESULTS

### Data characteristics and distribution

A total of 11 data sets from 10 independent studies and one public database, all utilizing 16S rRNA gene sequencing, were included in the final analysis. Technical batch effects across these combined data sets were rigorously corrected prior to downstream analysis, resulting in a significant reduction in Bray-Curtis distances and diminished technical variation across batches (Fig. S1). This comprised 76 URTs (oral wash, nasal, and oropharyngeal swabs), 269 sputum samples, and 73 BALF samples from PTB patients (Table 1). These specimens were sourced from a total of 14 countries. The largest proportion of samples originated from South Africa ($n = 120$, including URTs and sputum samples), followed by China ($n = 73$, including sputum and BALF samples). For analytical purposes, countries contributing 10 or fewer samples were grouped into an "Other" category. Additionally, there were 17 URTs, 46 sputum specimens, 13 BALF samples from the HC group, which were collected from subjects in four countries (Table S1).

### Distinct microbial community structures across PTB patients' respiratory specimens

At the population level, taxonomic profiling revealed distinct genus-level community structures across respiratory sample types. BALF samples harbored a greater number of genera compared to URTs and sputum, with 20.8% of genera shared among all three types (Fig. 2A). Alpha diversity analysis demonstrated that URTs consistently exhibited the lowest median diversity indices, whereas sputum samples showed intermediate diversity levels. Notably, BALF samples showed the highest α-diversity, with significantly elevated Chao1 and ACE indices (Fig. 2B). Then, PCoA was performed to evaluate overall differences in microbial community composition. The results revealed a clear separation of URTs and sputum communities from BALF samples. Although URTs and BALF communities were distinct, they exhibited partial overlap (Fig. 2C). We next compared microbial diversity between PTB and the HC group within each respiratory sample type. The results showed that the alpha diversity indices of all three sample types in the PTB group were significantly lower than those in the HC group (Fig. S2A). Consistently, PCoA showed that microbial community composition in the PTB was significantly distinct from that of HC across respiratory niches (Fig. S2B).

Analysis of the top 20 genera across samples revealed distinct microbial features. *Streptococcus*, *Prevotella*, *Veillonella*, and *Neisseria* were consistently abundant in all three sample types. However, distinct signature genera were identified in specific sample types: URTs were characterized by relatively high abundances of *Rothia* (Fig. 2D), *Roseateles* was exclusively detected among the top 20 genera in URT and sputum samples and exhibited relatively consistent abundance across different countries (Fig. 2E). Notably, BALF microbiota contained several genera less prominent in other samples, including *Mycobacterium* and *Serratia;* the *Serratia* was enriched in the samples from China (Fig. 2F).

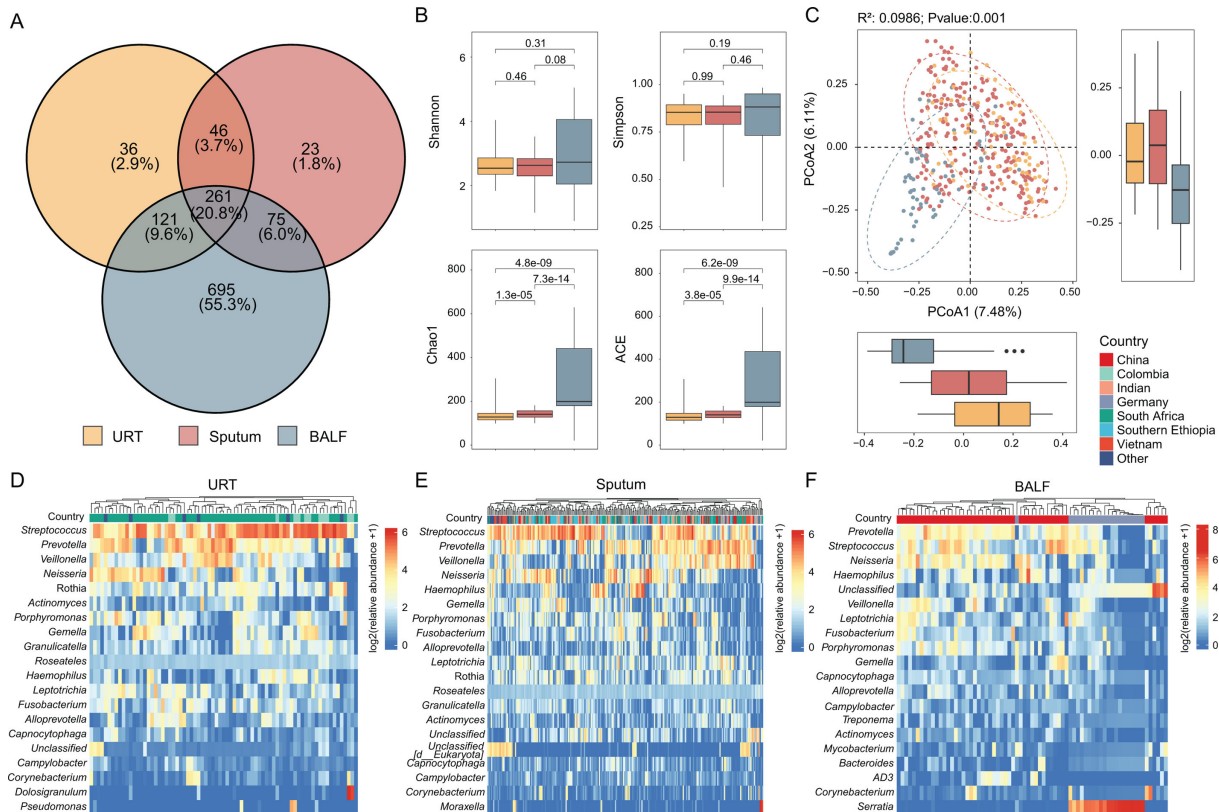

**FIG 2** Microbial community diversity and composition across respiratory sample types. (A) Venn diagram showing the number of unique and shared bacterial genera among the three sample types. (B) Alpha diversity indices (Simpson, ACE, Shannon, and Chao1) for URTs, sputum, and BALF samples. Boxplots display the median and interquartile range for each metric (significant difference: $P < 0.05$). (C) PCoA plot based on Bray-Curtis distance, the statistical significance of the separation was tested using PERMANOVA, with $R^2$ and $P$ values shown. (D–F) Heatmaps displaying the relative abundance (log2(abundance+1)) of the top 20 most abundant genera in (D) URTs, (E) sputum, and (F) BALF samples. Samples are grouped by country of origin (Germany, China, Colombia, India, Ethiopia, Vietnam, South Africa, and an "Other" group).

## Microbial community variations across sample types and different groups

To further elucidate the microbial distribution differences among the three sample types, we identified differentially abundant genera with LEfSe analysis. The results revealed four genera enriched in URTs, including *Streptococcus*, *Rothia*, *Granulicatella*, and *Gemella*. In sputum samples, *Unclassified_[d__Eukaryota]*, *Prevotella*, and *Veillonella* were identified. For BALF samples, *Serratia* and *Unclassified_[Unassigned]* were found to be differentially abundant (Fig. 3A). Intergroup comparisons further showed that *Serratia* was significantly higher in BALF than in URTs or sputum, while *Unclassified_[d__Eukaryota]* was significantly lower in URTs. The *Streptococcus* exhibited significant differences across all three sample types, and *Gemella* showed a significant difference only between sputum and BALF samples. No significant difference was found for *Prevotella* across sample types (Fig. 3B). Among them, *Serratia* was almost exclusively detected in BALF samples from Germany, while *Unclassified_[d__Eukaryota]* was found almost solely in sputum samples (Fig. 3C).

A comparative analysis of differential bacterial genera between the PTB and HC groups across sample types was next conducted. The results showed that *Streptococcus* and *Neisseria* were relatively higher in URTs from the PTB group (Fig. S3A). In sputum samples, *Streptococcus* and *Unclassified_d_Eukaryota* were enriched in the PTB group (Fig. S3B), whereas *Streptococcus* was more abundant in BALF samples from the HC group (Fig. S3C). Additionally, *Fusobacterium* levels were higher in both URTs and sputum samples from the HC group (Fig. S3A and B), but showed the opposite trend in BALF samples, with higher levels observed in PTB (Fig. S3C).

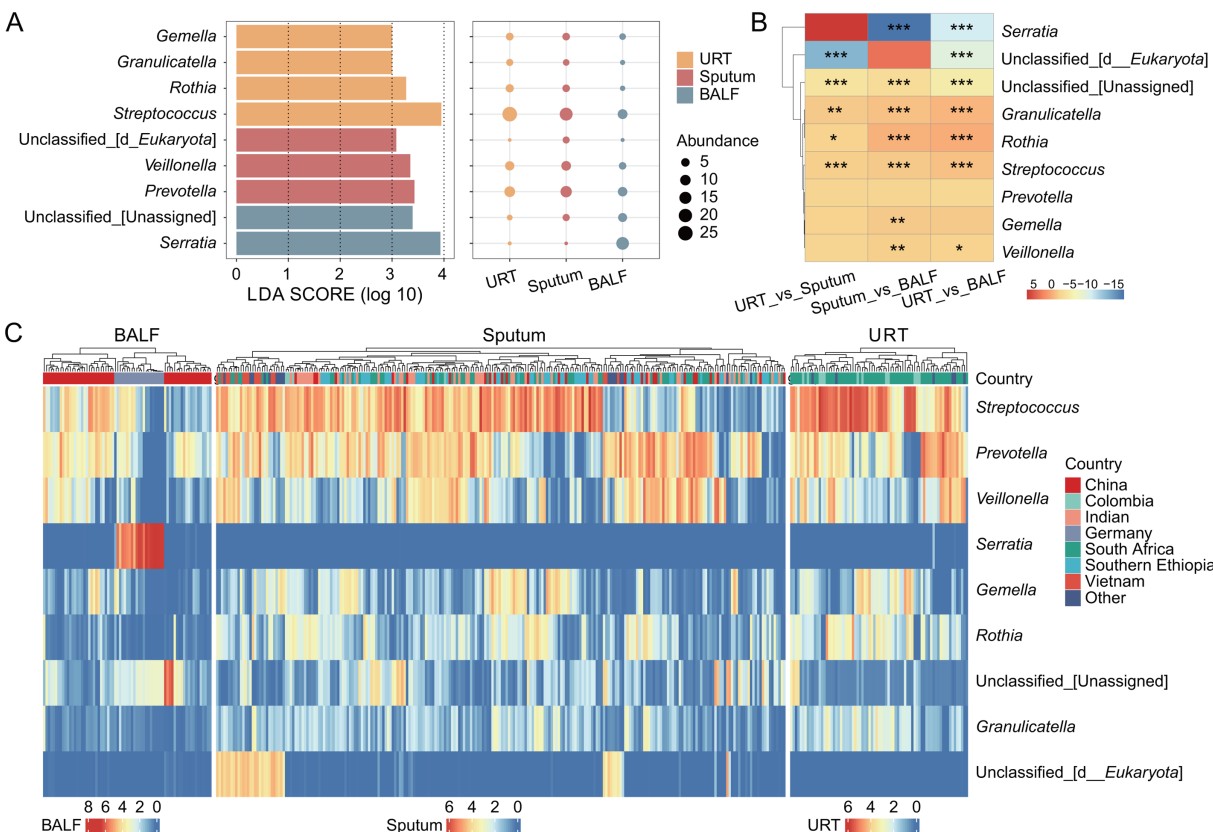

FIG 3 Microbial community composition and differential abundance of sample types with PTB. (A) LEfSe results identifying taxa with statistically significant differences in abundance among URTs, sputum, and BALF samples. The histogram shows the LDA scores (log10) for taxa enriched in each specific sample type, with only scores exceeding a significant threshold displayed. (B) Pairwise comparisons among all three types were performed using the rank-sum test. The color of each data point represents the fold change in abundance between the compared groups. Statistical significance is denoted by asterisks: *$P < 0.05$, **$P < 0.01$, ***$P < 0.001$. (C) Heatmap of relative abundance of differentially abundant genera across sample types.

## Microbial network structure and community assembly patterns in PTB

At the population level, co-occurrence network analysis revealed distinct patterns of microbial associations across respiratory sample types in PTB. The URT network exhibited a moderate structure with 37 nodes and 166 edges, dominated by *Firmicutes* (35.14%) and *Bacteroidota* (18.92%), where positive interactions (58.43%) slightly prevailed (Fig. 4A). In contrast, the sputum network was markedly more complex and interconnected, comprising 70 nodes and 623 edges with a *Firmicutes* predominance (35.71%), and displayed a stronger preponderance of positive correlations (64.21%) (Fig. 4B). The BALF network presented an intermediate complexity with 47 nodes and 318 edges, also primarily composed of *Firmicutes* (38.30%). Notably, its positive (50.31%) and negative (49.69%) correlations were nearly balanced (Fig. 4C).

To assess the structural robustness of these inferred networks, we simulated species loss via robustness analyses. The robustness curves of the three networks largely overlapped (Fig. 4D), suggesting comparable structural sensitivity to perturbation at the population level, despite differences in network complexity. All networks showed a gradual decline in connectivity under random node removal, but a sharp drop when high-degree nodes were targeted (Fig. 4D), indicating a common structural reliance on well-connected taxa.

We further applied a neutral community model to evaluate the relative contribution of stochastic processes to microbial community assembly across respiratory niches. The model fit ($R^2$) was highest for sputum ($R^2 = 0.6078$, $m = 0.3506$), followed by URTs ($R^2 = 0.4765$, $m = 0.1173$) and BALF ($R^2 = 0.3449$, $m = 0.0252$) (Fig. 4E). These results suggest

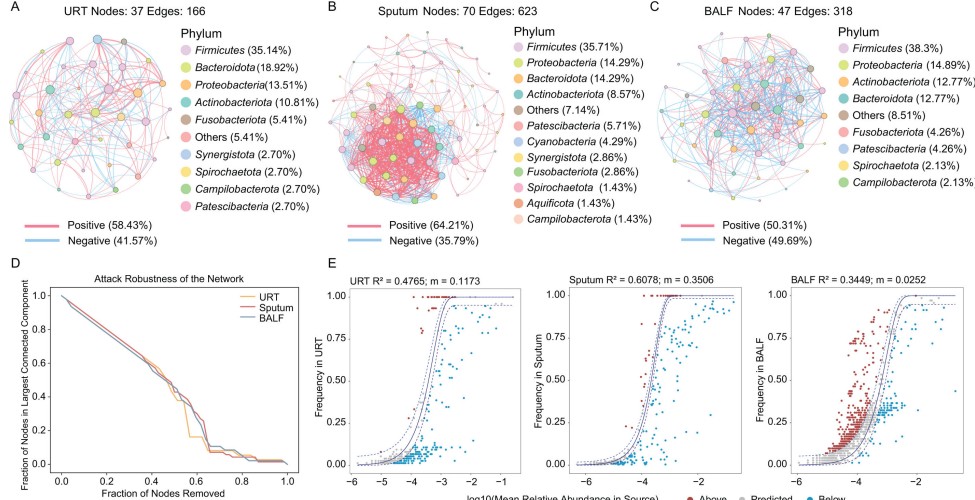

**FIG 4** Microbial co-occurrence network topology, robustness, and neutral community model analysis. (A–C) Microbial co-occurrence networks for (A) URTs, (B) sputum, and (C) BALF samples. Nodes represent bacterial genera, colored by phylum affiliation. Edges represent significant correlations. Pie charts indicate the proportion of positive and negative interactions within each network. The total number of nodes and edges for each network is displayed. (D) Network robustness is evaluated by simulating random node removal. The curve shows the fraction of nodes remaining in the largest connected component as nodes are progressively removed. (E) Neutral model fit for each sample type. The solid line represents the model prediction, and the dashed lines indicate the 95% confidence interval. The coefficient of determination ($R^2$) indicates the goodness-of-fit to the neutral model, and the migration rate (M) reflects the estimated dispersal rate for each community.

that random processes contribute more prominently to the assembly of sputum-associated communities at the population level. In contrast, the lower fit for BALF implies a relatively greater influence of deterministic processes.

## PTB microbiome functional profiling predicts niche-specific metabolic characteristics

In the overall population, to further explore the predicted functional potential of the respiratory microbiome, KEGG pathways and KO functional profiles were inferred. Among the top 50 predicted KEGG pathways shared across the three respiratory sample types, peptidoglycan maturation was the most abundant pathway, followed by the adenosine and adenosine salvage III pathway (Fig. 5A). Both pathways showed higher predicted abundances in sputum and URT samples (Fig. S4). Similarly, the superpathway of tetrahydrofolate biosynthesis and salvage was predicted to be more abundant in sputum and URT samples (Fig. S4). Overall, predicted pathway profiles were highly similar between URT and sputum samples (Fig. 5A), with few statistically significant differences detected (Fig. S4). Moreover, predicted functional category abundances showed minimal variation across geographic origins within each sample type (Fig. 5A).

Subsequent correlation analysis between the top 50 pathways and bacterial genera was performed, restricted to pathways documented in the MetaCyc metabolic database. Significant correlations were observed between 24 genera and 15 predicted functional pathways. Among these, the UDP-N-acetylmuramoyl-pentapeptide biosynthesis II demonstrated a significantly positive correlation with *Streptococcus*, while showing negative correlations with genera such as *Pseudarthrobacter*, *Enterococcus*, and *Staphylococcus*. The peptidoglycan biosynthesis I pathway was significantly negatively correlated with *Pseudomonas*, with similar relationships observed for *Geobacillus* and *Sinomonas*. Additionally, *Bacillus* exhibited positive correlations with multiple pathways, including UMP biosynthesis III, 5-aminoimidazole ribonucleotide biosynthesis I, and chorismate biosynthesis (Fig. 5B).

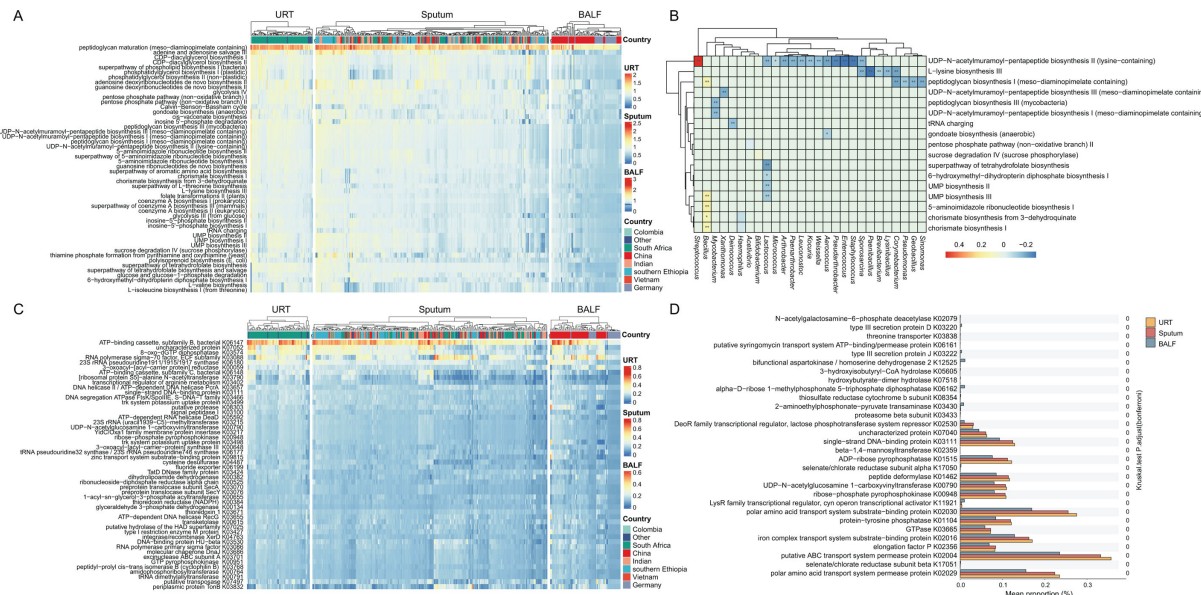

**FIG 5** Functional profiling and taxonomic associations of the respiratory microbiome. (A) A heatmap displaying the relative abundance (mean proportion, %) of the 50 most enriched microbial metabolic pathways. (B) A correlation heatmap illustrating the relationships between significantly enriched metabolic pathways and dominant bacterial genera. The color scale represents the strength and direction of the association (*$P < 0.05$, **$P < 0.01$, ***$P < 0.001$). (C) The heatmap shows the relative abundance of the 50 most enriched gene functions, identified by their K numbers. (D) The KOs that are significantly differentially abundant between pre-defined sample groups. Statistical significance was determined using the Kruskal-Wallis test with Bonferroni correction for multiple comparisons.

Furthermore, we identified the top 50 predicted KO functions across the three sample types. ATP-binding cassette subfamily B (bacterial) was the most abundant predicted function, followed by uncharacterized protein, 8-oxo-dGTP diphosphatase, and RNA polymerase sigma-70 factor. Variation in predicted KO abundances across geographic origins was observed for a subset of functions (Fig. 5C). In addition to the KO functions mentioned above, URTs showed minimal geographic variation in functional levels. In sputum samples, the abundance of K06148, K03790, and K03420 fluctuated depending on the region, while the remaining functional profiles remained largely consistent. German BALF samples displayed overall lower predicted functional levels (Fig. 5C). Finally, differential analysis of the enriched KO functions revealed the top 30 significantly different functions. Notably, putative ABC transport system permease protein K02004 and polar amino acid transport system protein (K02030, K02029) were significantly higher in URTs and sputum samples compared to the BALF. Additionally, bifunctional aspartokinase/homoserine dehydrogenase K12525 and alpha-D-ribose 1-methylphosphonate 5-triphosphate diphosphatase K06162 are predicted to be more abundant in BALF samples (Fig. 5D).

## DISCUSSION

This study conducted, for the first time, a microbiome meta-analysis of three respiratory tract samples from patients with pulmonary tuberculosis alongside relevant healthy controls, aiming to systematically reveal the characteristics of their community structure. Although the samples we screened come from different populations and regions, to a certain extent, they can help us gain a deeper understanding of the differences in the microbial communities of samples from different respiratory tract sites under the state of pulmonary tuberculosis.

Our findings demonstrated that BALF samples displayed higher alpha diversity compared to other samples (Fig. 2B), a pattern consistent with earlier reports characterizing site-specific microbial variation across respiratory niches (26). This reinforces the influence of sampling location on not only diversity but also the overall community structure in the respiratory tract of PTB patients (Fig. 2C). The partial overlap between

URT and lower respiratory samples in PCoA may indicate microbial aspiration from upper airways—a phenomenon noted in prior studies (23). Notably, samples originated from different individuals, and geographic or demographic factors may also contribute to variation. Compared to the HC group, PTB patients exhibited reduced alpha diversity across all sample types (Fig. S2A), consistent with known diversity loss in disease (27, 28). Furthermore, the microbial community composition of the PTB group was significantly distinct from that of the HC (Fig. S2A). Together, these findings indicate that PTB is associated with notable alterations in respiratory tract microbiota.

Despite variations in microbial diversity, the three sample types harbor a common foundational microbiota, including *Streptococcus*, *Prevotella*, *Veillonella,* and *Neisseria* (Fig. 2D through F), which is consistent with a previous study (29). The unique enrichment patterns in different parts highlight the key influence of the local microenvironment. Notably, *Serratia* was significantly enriched in German BALF samples (Fig. 2F and 3B). This genus is known to produce various glycolipids and lipopeptides with documented antimicrobial, antifungal, and antiprotozoal activities (30), which may account for the generally weaker enrichment of top 20 species in German BALF samples compared to Chinese counterparts.

Differential species analysis revealed distinct microbial signatures across respiratory niches (Fig. 3A and B). *Streptococcus* was consistently enriched in the URTs and sputum of PTB patients (Fig. 3A), and its abundance in URTs was significantly higher than in HC samples (Fig. S3A). This enrichment is consistent with previous reports describing Streptococcus as an opportunistic pathobiont associated with inflammatory airway conditions, rather than direct evidence of within-host proliferation or pathogenic activity in PTB (31, 32). *Rothia* enrichment in URT samples is consistent with its role as an upper respiratory tract commensal but should be interpreted cautiously given its opportunistic pathogenic potential (33). Although *Fusobacterium* has been implicated in multiple inflammatory and infectious diseases (34), its enrichment in PTB should be interpreted as a population-level association rather than a causal relationship, warranting further investigation with paired sampling and functional validation.

Co-occurrence networks revealed niche-specific dysbiosis in PTB, while all networks exhibited similar fragility upon hub taxon removal. Meanwhile, neutral modeling further suggested that these distinct network structures may be linked to differing ecological assembly processes across respiratory compartments. Specifically, the microbial network in sputum samples displayed higher complexity (Fig. 4B), with stochastic processes likely playing a dominant role in its community assembly (Fig. 4E). This may be attributed to its origin in the lower respiratory tract, combined with potential upper airway contamination, collectively fostering a microbial community characterized by dense and diverse interspecies interactions. In contrast, the interaction network in BALF samples exhibited a relatively balanced ratio of positive to negative correlations (Fig. 4C), while neutral community modeling further indicated a lesser influence of stochastic processes on its composition (Fig. 4E). URT samples showed intermediate network and assembly characteristics (Fig. 4E), which may reflect their exposure to environmental perturbations at the population level, consistent with the role of the upper respiratory tract as an ecological "gateway" (10). Furthermore, these inferred interactions necessitate validation through future experimental or longitudinal studies.

Functional prediction revealed distinct metabolic characteristics across respiratory niches. The widespread predicted enrichment of the peptidoglycan maturation pathway (Fig. 5A) suggests a potentially increased capacity for bacterial cell wall biosynthesis, which may support microbial survival and adaptability in the host environment (35). The predicted enrichment of the adenine and adenosine salvage III pathway in URT and sputum samples (Fig. S4) is consistent with a hypothesized state of energy stress among microorganisms in these niches (36, 37). Furthermore, a positive correlation was observed between *Streptococcus* abundance and the predicted potential for a key peptidoglycan biosynthesis step (Fig. 5B). While the correlations between specific taxa and predicted pathways (e.g., *Streptococcus* with peptidoglycan biosynthesis) generate

biologically plausible hypotheses, they do not establish causality or confirm *in vivo* function. Additionally, the multiple positive correlations of *Bacillus* with biosynthesis pathways suggest it may be a metabolically versatile actor in this ecosystem (Fig. 5B).

Finally, KO functional analysis revealed that the ATP-binding cassette (ABC) transporter system (K06147) was highly abundant (Fig. 5C). This enriched function, which was particularly significant in URT and sputum samples (Fig. 5D), may indicate an increased potential for substrate transport relevant to cell wall biosynthesis, including pathways such as peptidoglycan maturation (38), and therefore warrants further investigation. Notably, geographic variation further complicated the predicted functional landscape, as evidenced by the lower predicted functional potential in German BALF samples (Fig. 5C). Moreover, most predicted functional pathways were significantly more abundant in URT and sputum samples than in BALF samples (Fig. 5D), suggesting greater functional plasticity at the population level outside the immediate pulmonary lesion microenvironment.

We acknowledge several limitations. While integrating multiple independent data sets increased statistical power and generalizability, it also introduced heterogeneity across sequencing platforms, sample processing, and clinical characteristics. Rigorous batch effect correction was therefore applied, preserving the overall biological interpretation while reducing technical noise; however, residual confounding from unmeasured or incompletely harmonized factors cannot be fully excluded. Furthermore, the inferred ecological interactions and predicted metabolic potential are inherently descriptive and hypothesis-generating, rather than direct evidence of causal mechanisms or *in vivo* biochemical activity. To advance these findings, future studies should employ prospective, multi-center designs with standardized protocols, paired longitudinal sampling from the same individuals to validate within-host dynamics, and multi-omics integration to functionally characterize active microbial processes in PTB.

## Conclusion

In conclusion, this study establishes a multi-site ecological framework of the respiratory microbiota in pulmonary tuberculosis, demonstrating that the respiratory tract functions as a spatially structured ecosystem rather than a homogeneous environment. The URT microbiota likely reflects environmental exposures and early colonization status. The sputum microbiota exhibits a stochastic assembly pattern, supporting its role as a mixing interface between the upper and lower respiratory tract. With its higher diversity, BALF is likely a more accurate proxy for host-microbial crosstalk in pulmonary lesions. Differences from healthy controls and site-specific ecological patterns advance the potential for precise microbial prediction and diagnosis of PTB. Collectively, this framework provides a foundation for niche-informed microbial biomarker development and advances efforts toward more precise and non-invasive tuberculosis diagnosis.

## ACKNOWLEDGMENTS

This research was funded by the National Natural Science Foundation of China (32394011), Beijing Nova Program (20250484980).

M.Q.: Writing—original draft, Investigation, Formal analysis, Data collection, Methodology, Conceptualization, Visualization. Y.W.: Writing—review & editing, Software, Methodology, Data curation, Formal analysis. S.L.: Writing—review & editing, Validation, Investigation, Visualization, Data curation. X.L., S.L., Y.L., and L.H.: Reviewing & editing, Validation, Software, Methodology, Data curation. H.X.: Reviewing, Validation, Software, Methodology, Investigation. L.L.: Writing—review & editing, Supervision, Project administration, Funding acquisition, Conceptualization. Y.P.: Writing—review & editing, Supervision, Project administration, Funding acquisition, Conceptualization.

## AUTHOR AFFILIATIONS

[1]Department of Epidemiology, School of Public Health, Cheeloo College of Medicine, Shandong University, Jinan, China

²Beijing Key Laboratory for Key Technologies in Tuberculosis Prevention and Control, Department of Bacteriology and Immunology, Beijing Chest Hospital, Capital Medical University/Beijing Tuberculosis and Thoracic Tumor Research Institute, Beijing, China
³Department of Scientific Affairs, Hugobiotech Co., Ltd., Beijing, China

## AUTHOR ORCIDs

Mingyang Qin  http://orcid.org/0000-0002-2010-905X
Yanhua Wen  http://orcid.org/0009-0002-6867-4602
Shanshan Li  http://orcid.org/0000-0003-4475-4536
Xuming Li  http://orcid.org/0000-0002-5049-9706
Han Xia  http://orcid.org/0000-0002-5179-0255
Yu Pang  http://orcid.org/0000-0001-6803-9807
Liang Li  http://orcid.org/0000-0001-5212-5890

## AUTHOR CONTRIBUTIONS

Mingyang Qin, Conceptualization, Formal analysis, Investigation, Methodology, Validation, Visualization, Writing – original draft | Yanhua Wen, Data curation, Formal analysis, Methodology, Software, Validation, Writing – review and editing | Shanshan Li, Conceptualization, Investigation, Validation, Writing – review and editing | Song Li, Data curation, Methodology, Software, Visualization, Writing – review and editing | Xuming Li, Methodology, Software, Validation, Writing – review and editing | Yuting Lin, Methodology, Software, Validation, Writing – review and editing | Long Hu, Methodology, Software, Validation, Writing – review and editing | Han Xia, Methodology, Software, Validation, Writing – review and editing | Yu Pang, Conceptualization, Funding acquisition, Investigation, Project administration, Supervision, Writing – review and editing | Liang Li, Conceptualization, Funding acquisition, Supervision, Validation

## DATA AVAILABILITY

All the data are derived from published articles and public databases.

## ADDITIONAL FILES

The following material is available online.

### Supplemental Material

**Supplemental Figures (mSystems01563-25-s0001.eps).** Figures S1-S4.
**Legends (mSystems01563-25-s0002.docx).** Supplemental figure legends.
**Table S1 (mSystems01563-25-s0003.docx).** Detailed data information of healthy control group.

### Open Peer Review

**PEER REVIEW HISTORY (review-history.pdf).** An accounting of the reviewer comments and feedback.

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
