## [Reviewer comments · mSystems]

The Respiratory Microbiome in Pulmonary Tuberculosis: A Meta-Analysis Reveals Niche-Specific Microbial and Functional Signatures

Mingyang Qin, Yanhua Wen, Shanshan Li, Song Li, Xu-ming Li, Yuting Lin, Long Hu, Han Xia, Yu Pang, and Liang Li

Corresponding Author(s): Yu Pang, 北京胸科医院

Review Timeline:

Submission Date:	November 4, 2025
Editorial Decision:	December 23, 2025
Revision Received:	February 6, 2026
Accepted:	February 16, 2026

Editor: Naseer Sangwan

Reviewer(s): Disclosure of reviewer identity is with reference to reviewer comments included in decision letter(s). The following individuals involved in review of your submission have agreed to reveal their identity: Simbarashe Peter Zvada (Reviewer #2)

Transaction Report:

DOI: <https://doi.org/10.1128/msystems.01563-25>

Re: mSystems01563-25 (**The Respiratory Microbiome in Pulmonary Tuberculosis: A Meta-Analysis Reveals Niche-Specific Microbial and Functional Signatures**)

Dear Prof. Yu Pang:

Revision Guidelines

Sincerely,
Naseer Sangwan
Editor
mSystems

Reviewer #1 (Comments for the Author):

This is a strong and solid contribution to be added to the area of PTB. The ideas provide solid addition to the area of microbiome biogeography with solid methods and ecological insights. It provides novelty and rigorous methodological approach that crystallized the ideas and the stated objectives.

Reviewer #2 (Comments for the Author):

Overall, this is a strong, data-rich meta-analysis with clear novelty, appropriate methodology, and a coherent ecological framework. Below are critiques and recommendations to authors for consideration.

1. Dataset heterogeneity and batch effects

The manuscript integrates multiple publicly available datasets generated using different sequencing platforms, 16S rRNA variable regions, and laboratory protocols. While the authors reprocessed all data using a unified pipeline, it remains unclear to what extent residual batch effects may influence the observed taxonomic and ecological differences across respiratory niches.

2. Functional inference based on 16S rRNA data

Functional interpretations rely on PICRUSt2 predictions derived from 16S rRNA gene data. These predictions reflect potential metabolic capacity rather than measured activity, and some interpretations appear stronger than warranted by the data.

3. Validation of the random forest classifier

Critique:

The random forest model demonstrates high discriminatory performance; however, model training and evaluation were conducted using cross-validation within the same pooled dataset. Without independent external validation, the generalizability of the classifier across cohorts and technical settings remains uncertain.

4. Interpretation of ecological modeling results

Critique:

Co-occurrence network analysis and neutral community modeling are correlational and descriptive approaches. Several statements in the Discussion suggest deterministic ecological mechanisms or causal relationships that are not directly supported by the modeling framework.

5. Lack of paired or longitudinal sampling

Samples from different respiratory compartments were not obtained from the same individuals. As a result, conclusions regarding microbial migration, niche adaptation, or spatial structuring should be interpreted as population-level associations rather than within-host processes.

Suggestions for Improvement:

1. Address heterogeneity explicitly

The authors should more clearly acknowledge inter-study heterogeneity as an inherent limitation of the meta-analysis and discuss how residual batch effects may influence results. Consideration of batch-aware statistical approaches or hierarchical modeling in future studies would strengthen the analytical framework.

2. Temper functional claims

Functional results should be consistently described as predicted or inferred metabolic potential, and conclusions should avoid implying direct biological activity. Explicit discussion of the limitations of 16S-based functional inference is recommended.

3. Clarify model scope and validation needs

Statements regarding the predictive or translational utility of the random forest classifier should be moderated. The authors should emphasize that external validation using independent cohorts is required to assess robustness and generalizability.

4. Refine ecological interpretation

Interpretations of network topology and neutral model results should be framed as tendencies or associations, not causal mechanisms. The authors are encouraged to highlight the need for experimental or longitudinal validation to confirm inferred ecological interactions.

5. Emphasize study design constraints

The manuscript should clearly state that comparisons across respiratory niches reflect population-level differences. Future directions should prioritize paired, multi-site, and longitudinal sampling designs to directly assess within-host spatial and temporal dynamics.

Minor comments:

- Consider adding control comparator from healthy population to the analysis to provide better comparison to the change in the microbiota across the different samples used.
- Since you have used 11 data sets in your analysis: consider accounting for the difference between the different datasets by adjusting for it or conducting sensitivity analysis providing that there is enough power for this.

Reviewer #1 (Comments for the Author):

This is a strong and solid contribution to be added to the area of PTB. The ideas provide solid addition to the area of microbiome biogeography with solid methods and ecological insights. It provides novelty and rigorous methodological approach that crystalized the ideas and the stated objectives.

We appreciate you for the encouraging and positive assessment of our work. We are pleased that the reviewer found the study to provide a rigorous methodological framework and meaningful ecological insights into PTB microbiome biogeography. Detailed information in our point-by-point responses. And in the revised manuscript, we have marked the modified parts in red for your convenience in viewing.

Minor comments:

- Consider adding control comparator from healthy population to the analysis to provide better comparison to the change in the microbiota across the different samples used.

Response:

We thank the you for this valuable suggestion. To ensure appropriate comparison and maximize comparability of control data, we systematically screened control samples across the 11 included datasets, encompassing healthy controls, non-tuberculosis disease controls, household contacts of tuberculosis patients, and individuals with latent tuberculosis infection. To minimize potential confounding and maintain cohort homogeneity, only samples explicitly defined as healthy controls (HC) were included in the final analysis (Page 5, Line 125-128).

Following strict screening, a total of 17 upper respiratory tract samples, 46 sputum samples, and 13 bronchoalveolar lavage fluid samples were retained (Page 8, Line 215-217). And the specific data information can be viewed in Table S1. Moreover, the results regarding diversity can be seen on pages 10, lines 238-243 (Figures S2). On page 12, lines 287 to 294, the results of the comparison of different species can be seen (Figures S3).

- Since you have used 11 data sets in your analysis: consider accounting for the difference between the deferent datasets by adjusting for it or conducting sensitivity analysis providing that there is enough power for this.

Response:

We sincerely thank you for this considerable suggestion. We agree that accounting for variability across datasets is critical for ensuring the robustness of meta-analytic findings. Accordingly, we adjusted for inter-dataset differences by applying batch-effect correction across all included datasets, with the dataset source explicitly modeled as the batch variable. The detailed methodology is described in the Methods section (Page 6, Line 161-170).

This correction substantially reduced dataset-driven technical variation, as illustrated in revised manuscript (Page 8, Line 206-209, Figure S1), while preserving biologically meaningful signals related to disease status and respiratory niche.

Although formal sensitivity analyses stratified by individual datasets were limited by sample size in certain subgroups, the consistency of key results before and after batch correction supports the robustness of our conclusions against dataset heterogeneity.

Reviewer #2 (Comments for the Author):

Overall, this is a strong, data-rich meta-analysis with clear novelty, appropriate methodology, and a coherent ecological framework. Below are critiques and recommendations to authors for consideration.

We are grateful for the positive evaluation of our study and for the constructive critiques and recommendations provided below. We have carefully considered each comment and revised the manuscript accordingly, as detailed in our point-by-point responses. And in the revised manuscript, we have marked the modified parts in red for your convenience in viewing.

1. Dataset heterogeneity and batch effects

The manuscript integrates multiple publicly available datasets generated using different sequencing platforms, 16S rRNA variable regions, and laboratory protocols. While the authors reprocessed all data using a unified pipeline, it remains unclear to what extent residual batch effects may influence the observed taxonomic and ecological differences across respiratory niches.

Response:

We sincerely appreciate the valuable feedback provided in your review. We fully agree with your concerns regarding potential batch effects. Accordingly, we have performed batch effect correction on the integrated data, and the detailed methodology has been added to the revised manuscript (Page 6, line 161-170). Thus, a reduction in discrepancies following batch effect correction was achieved (Page 8, Line 206-209). Additionally, a further explanation regarding this approach is provided in our response to your subsequent suggestions.

2. Functional inference based on 16S rRNA data

Functional interpretations rely on PICRUSt2 predictions derived from 16S rRNA gene data. These predictions reflect potential metabolic capacity rather than measured activity, and some interpretations appear stronger than warranted by the data.

Response:

We agree that PICRUSt2-based functional predictions reflect potential metabolic capacity rather than measured activity. And we have revised the manuscript to temper functional interpretations and explicitly state this limitation, framing these results as hypothesis-generating. The specific modifications can be viewed in the results section at line 352 to 390 of the original text.

3. Validation of the random forest classifier

Critique:

The random forest model demonstrates high discriminatory performance; however, model training and evaluation were conducted using cross-validation within the same pooled dataset. Without independent external validation, the generalizability of the classifier across cohorts and technical settings remains uncertain.

Response:

We thank you for this important comment and appreciate the opportunity to clarify our modeling strategy. We indeed did not conduct independent external validation. However, based on your suggestions, we incorporated independent external validation in the subsequent analysis. For the specific content, please refer to the subsequent improvement suggestions section, where detailed answers are provided regarding the opinions related to the random forest model.

4. Interpretation of ecological modeling results

Critique:

Co-occurrence network analysis and neutral community modeling are correlational and descriptive approaches. Several statements in the Discussion suggest deterministic ecological mechanisms or causal relationships that are not directly supported by the modeling framework.

Response:

Thank you for your insightful comment. We agree that these analyses are correlational, and we have revised the manuscript to avoid causal or deterministic interpretations. The specific modifications can be viewed in the results section at page 14-15, line 312 to 336 of the original text.

5. Lack of paired or longitudinal sampling

Samples from different respiratory compartments were not obtained from the same individuals. As a result, conclusions regarding microbial migration, niche adaptation, or spatial structuring should be interpreted as population-level associations rather than within-host processes.

Response:

We feel great thanks for highlighting this important limitation. We agree that, because samples from different respiratory compartments were obtained from different individuals, our analyses reflect population-level patterns rather than within-host microbial dynamics. And we have made some revisions to the description of this aspect in the article. As detailed in line and 313 and 352 “At the population level...” “In the overall population...”

Suggestions for Improvement:

1. Address heterogeneity explicitly

The authors should more clearly acknowledge inter-study heterogeneity as an inherent

limitation of the meta-analysis and discuss how residual batch effects may influence results. Consideration of batch-aware statistical approaches or hierarchical modeling in future studies would strengthen the analytical framework.

Response:

We agree that integrating publicly available datasets generated using different sequencing platforms, 16S rRNA regions, and laboratory protocols inevitably introduces inter-study heterogeneity. To address this, all raw data were reprocessed using a unified bioinformatics pipeline, and batch-effect correction was applied with the dataset source explicitly modeled as the batch variable while retaining disease state and sample type as biological covariates. Importantly, the major taxonomic patterns and ecological interpretations were consistent before and after batch-effect correction, indicating that the observed differences across respiratory niches are unlikely to be driven by batch-specific artifacts. Nevertheless, we acknowledge that residual batch effects from unmeasured or incompletely harmonized factors may persist, and this limitation has now been explicitly noted in the revised manuscript (Page 21, line 495-498). As shown in the figure below, the batch effects were substantially reduced after data correction.

We have further discussed that, despite rigorous batch-effect correction and unified data processing, residual batch effects and unmeasured confounding factors may still influence certain taxonomic and ecological inferences. These revisions improve the transparency of the analytical framework while maintaining the robustness of the primary conclusions.

2. Temper functional claims

Functional results should be consistently described as predicted or inferred metabolic potential, and conclusions should avoid implying direct biological activity. Explicit discussion of the limitations of 16S-based functional inference is recommended.

Response:

We think this is an excellent suggestion. In the revised manuscript, all functional results derived from 16S rRNA data are now consistently described as predicted or inferred metabolic potential, and language implying direct biological activity has been removed (Page 15-17, Line 352-390). We have also explicitly expanded the Discussion to address the limitations of 16S-based functional inference, emphasizing

that these predictions are hypothesis-generating rather than evidence of *in vivo* metabolic activity (Page 21, line 498-500). These revisions ensure that functional interpretations are appropriately cautious and aligned with the underlying data.

3. Clarify model scope and validation needs

Statements regarding the predictive or translational utility of the random forest classifier should be moderated. The authors should emphasize that external validation using independent cohorts is required to assess robustness and generalizability.

Response:

We fully agree with your point and will explain it in detail. Specifically, all samples were randomly split into a training set (70%) and an independent test set (30%). The training set was used to identify key features and to train the random forest model using the *Random Forest Classifier* module implemented in the Scikit-learn Python package. Feature importance was ranked based on the trained model, and the top 10 most informative features were subsequently used to construct a multiclass prediction model. The independent test set, which was not involved in either feature selection or model training, was used exclusively to evaluate model performance.

The ROC curves derived from the independent test set are shown in Figure A for the uncorrected dataset and in Figure B for the dataset after batch-effect correction. Notably, random forest models built using either the original data or the batch-corrected data consistently demonstrated good predictive performance on the independent test set, supporting the stability of the selected features and the robustness of the classification results.

We would like to clarify that the datasets included in this study exhibit strong sample-type specificity, as most individual datasets contain only a single respiratory sample type. Under this circumstance, batch variables and biological covariates—particularly sample type—are highly collinear.

To address this challenge, we selected ConQuR, a batch-correction method specifically designed to be robust to collinearity between batch effects and covariates. In our analysis, ConQuR was applied with disease status and sample type specified as covariates and dataset source defined as the batch variable. This strategy was chosen to minimize technical variability while preserving biological signals associated with disease status.

However, because sample type was included as a covariate during batch

correction, the corrected data may inadvertently amplify sample-type-associated signals in a manner that is not appropriate for downstream supervised learning. In particular, using such corrected data for random forest classification could introduce bias and artificially inflate predictive performance. To avoid potential overinterpretation and ensure methodological rigor, we therefore chose not to apply the batch-corrected data to random forest modeling.

In light of these considerations, and to maintain a conservative and transparent analytical framework, we ultimately removed the random forest prediction results from the revised manuscript. We believe this decision strengthens the overall rigor of the study by avoiding model assumptions that may not be fully supported by the structure of the available data.

4. Refine ecological interpretation

Interpretations of network topology and neutral model results should be framed as tendencies or associations, not causal mechanisms. The authors are encouraged to highlight the need for experimental or longitudinal validation to confirm inferred ecological interactions.

Response:

We feel great thanks for your professional suggestions. In the revised manuscript, interpretations of co-occurrence network topology and neutral model results are consistently framed as tendencies or population-level associations rather than causal ecological mechanisms (Page 14-15, line 312-336). We have also explicitly emphasized the need for experimental, paired, and longitudinal studies to validate inferred ecological interactions and to directly assess underlying ecological processes (Page 19-20 line 452-467, Page 21, line 500-504).

5. Emphasize study design constraints

The manuscript should clearly state that comparisons across respiratory niches reflect population-level differences. Future directions should prioritize paired, multi-site, and longitudinal sampling designs to directly assess within-host spatial and temporal dynamics.

Response:

We sincerely thank the important suggestion. We agree that, because samples from different respiratory niches were not obtained from the same individuals, all cross-niche comparisons in this study reflect population-level differences rather than within-host spatial or temporal dynamics.

In the revised manuscript, we have explicitly clarified this design constraint throughout the results and discussion and have tempered interpretations accordingly. We have also expanded the discussion to emphasize that future studies incorporating paired, multi-site, and longitudinal sampling designs will be essential to directly resolve within-host microbial spatial organization and temporal dynamics (Page 21, line 492-504).

Re: mSystems01563-25R1 (**The Respiratory Microbiome in Pulmonary Tuberculosis: A Meta-Analysis Reveals Niche-Specific Microbial and Functional Signatures**)

Dear Prof. Yu Pang:

Your manuscript has been accepted, and I am forwarding it to the ASM production staff for publication. Your paper will first be checked to make sure all elements meet the technical requirements. ASM staff will contact you if anything needs to be revised before copyediting and production can begin. Otherwise, you will be notified when your proofs are ready to be viewed.

Sincerely,
Naseer Sangwan
Editor
mSystems

Reviewer #1 (Comments for the Author):

the response by the authors was satisfactory, this is great work

Reviewer #2 (Comments for the Author):

All my comments were adequately addressed. This work is an important addition to Scientific Community and TB treatment Consortium.